# The Principle of Action and Reaction According to Newton

Danilo Capecchi 

Structural and Geotechnical Engineering, University "La Sapienza", I-00184 Rome, Italy; danilo.capecchi@uniroma1.it; Tel.: +39-(0)-644585085

**Definition:** The principle of action and reaction is generally considered the least problematic and interesting of Newton's three laws of dynamics—least problematic because it seems self-evident, and least interesting because Newton's mechanics of *Principia* essentially represents the dynamics of a mass point, while the principle of action and reaction is mainly important in the case of a set of bodies that interact with each other. However, reading Newton's text is enough for the principle to appear equally problematic and interesting as the other two. This entry aims to justify this statement and to help clarify the meaning of the principle.

**Keywords:** dynamics; natural philosophy; principle of action and reaction; Newton; laws of motion



## 1. Introduction

Newton's mechanics, which was not only presented in the *Principia*, has been the subject of a huge number of writings by historians of science and beyond. Particular attention has been paid to the first two laws of motion, whereas there has been less interest in the third law, known as the principle of action and reaction (hereinafter PA) [1–7]. The reasons could be, on the one hand, the apparent evidence for the principle itself, (There are some difficulties in having a complete understanding of its meaning in pedagogical situations [8–11]) and, on the other hand, the modest use that Newton made of it in the *Principia* given that their own mechanics, more precisely its theoretical core, is essentially that of the mass point, while PA is mainly important in the study of systems of interacting bodies.

PA has been studied by philosophers of nature and by mathematicians from different points of view. Philosophers have been interested in its ontological and epistemological nature and have often treated it as a metaphysical principle, namely that every action corresponds to a reaction. They have all considered it a principle whose logical status is not very different from that according to which to each cause corresponds an effect. Mathematicians, more precisely mechanics scholars, started from everyday experience. According to them, for example, when two people play tug of war, it is evident that the pulling of one side corresponds to the pulling of the other. However, it is only in the 17th century with Newton that the principle reached a form that allowed for its application in a physical mathematical theory, specifying that the action and reaction have the same nature, are measurable entities, and are equal and contrary.

Ernst Mach gave great relevance to this principle, by asserting, for instance "Perhaps the most important achievement of Newton with respect to the principles is the distinct and general formulation of the law of the equality of action and reaction, of pressure and counter-pressure" [12] (p. 198). For him, it is the basis of a new formulation of mechanics. However, even Newton had to give great importance to the principle because, in the *Principia*, he attributed the first two laws of motion to Galileo Galilei ("Galileus invenit" [13] (p. 21) but he gave no attribution to the third law.

In this work, I will show how PA is not as simple as it seems, and in any case, it is no longer easier to frame than the other two principles/laws of dynamics; indeed, from certain points of view, it is the most complex because it is discussed here in terms of the ontology of the force.

## 2. The Principle of Action and Reaction in the *Principia*

A modern formulation of the principle of action and reaction in classical mechanics states that the application of a force to a body results in an equal and contrary reaction—in the same straight line—acting on another body. To put it differently, nature shows a basic symmetry by which forces go in couples with equal and contrary members and act only through bodies, either by contact or at a distance.

It is hard to say, and no attempt will be made in the following to distinguish to what extent the principle expresses an empirical truth or a mental construct. In some situations, it can be confirmed empirically (see below). In classical mechanics, only apparent violations of the principle are registered, which disappear with careful scrutiny [14–17]. The situation is different in the case of relativistic or quantum mechanics, where its validity is not clear and in some situations is also questioned [18–20].

Newton's original formulation of the principle and its commentary read as follows:

> **Law 3.** To any action there is always an opposite and equal reaction; in other words, the actions of two bodies upon each other are always equal and always opposite in direction.
> Whatever presses or draws something else is pressed or drawn just as much by it. If anyone presses a stone with a finger, the finger is also pressed by the stone. If a horse draws a stone tied to a rope, the horse will (so to speak) also be drawn back equally toward the stone, for the rope, stretched out at both ends, will urge the horse toward the stone and the stone toward the horse by one and the same endeavor to go slack and will impede the forward motion of the one as much as it promotes the forward motion of the other. If some body impinging upon another body changes the motion of that body in any way by its own force, then, by the force of the other body (*because of the equality of their mutual pressure*) [emphasis added] , it also will, in turn, undergo the same change in its own motion in the opposite direction. By means of these actions, equal changes occur in the motions, not in the velocities—that is, of course, if the bodies are not impeded by anything else. For the changes in velocities that, likewise, occur in opposite directions are inversely proportional to the bodies because the motions are changed equally. This law is valid also for attractions, as will be proved in the next scholium [13] (p. 14. Translation into English in [21]).

A modern reader immediately notices a particularity of the statement: there is no explicit reference to forces. Newton spoke of action and not of force, and in the example following the statement of the principle, it is clear that the idea of action is not necessarily that of force. He refers to three sources of action, pressure, impact, and attraction, which are in fact the same instances that he had qualified as sources of impressed forces in Definition 4 of the *Principia*, "Moreover, there are various sources of impressed force, such as percussion, pressure, or centripetal force" [13] (p. 2).

Indeed, the modern reader recognizes in pressure and attraction two sources of forces in the current sense. However, the identification of the action due to impact with a force looks different. Additionally, Newton did not help very much when he spoke of the forces of bodies in motion. Indeed, in the mechanics of the *Principia*, there is no room for this kind of force. Therefore, how can Newton's words be understood?

Newton's words suggest that action and reaction in the impact correspond to two impressed forces exchanged by the impacting bodies, which are declared to be equal and contrary. This equality can be justified into two ways, which both call for the second law of motion. "*Law* 2. A change in motion is proportional to the motive force impressed and takes place along the straight line in which that force is impressed" [13] (p. 13. Translation into English in [21]).

The first kind of justification is based on an experimental truth. In the impact of two bodies A and B, their overall momentum does not change because of the impact. However,

for Law 2 of motion, the immutability of momentum implies the equality of the force impressed by A onto B and that impressed by B onto A, with the sign reversed. The second kind of justification is suggested by the quite concise note by Newton: "(because of the equality of their mutual pressure)". That is, the force actually acting in the impact is the mutual pressure between the two impacting bodies. As the pressures are equal (and contrary), the impressed forces and consequently the variation in momenta are equal and contrary.

The fact that it is pressure that is responsible for the change in the momentum of impacting bodies is clearly expressed in the statement of the third law of motion in *De motu corporum in medijs regulariter cedentibus*, written before the *Principia*, possibly in 1684.

> As much as any body acts on another so much does it experience in reaction. *Whatever presses or pulls another thing by this equally is pressed or pulled* (The evidenced statement can be found also in the Lucasian *Lectures on algebra* [22], vol. 5, Lect. 6 [1675?], p. 148). If a bladder full of air presses or carries another equal to itself both yield equally inwards. If a body impinging on another changes by its force the motion of the other then its own motion (by reason of the equality of the mutual pressure) will be changed by the same amount by the force of the other [23] (Translation into English in [24]).

Here, the reference to two bladders filled with air suggests the mechanism of action and reaction, the exchange of equal and contrary pressures.

However, the role of pressure is still more clear in the dynamic sections of the *Waste book* dating from 1664. In axiom 8, the case of an impact of hard (plastic) bodies is considered:

> 8. If two quantitys (*a* and *b*) move towards one another and meete in <illeg> o. Then ye difference of theire motion shall not bee lost nor <illeg> loose its determination. For at their *occursion they presse equally uppon one another* [emphasis added], [and] therefore one must loose noe more motion yn ye other doth; soe yt ye difference of their motions cannot be destroyed [25] (f. 11r. Diplomatic transcription).

While axiom 9 refers to the impact of two elastic bodies:

> 9. If two equall and equally swift bodys (*d* and *c*) meete one another they shall bee reflected, soe as to move as swiftly frome one another after the reflection as they did to one another before it. For first suppose the sphaericall bodys *e*, *f* to have a springing or elastic force soe that meeting one another they will relent and be pressed into a sphaeroidicall figure, and in that moment in which there is a period put to theire motion towards one another theire figure will be most sphaeroidical and theire pression one upon the other is at the greatest, and if the endeavour to restore theire sphaericall figure bee as much vigorous and forcible as theire pressure upon one another was to destroy it they will gaine as much motion from one another after their parting as they had towards one another before theire reflection [...] *and there cannot bee succeede divers degrees of pressure twixt two bodys in one moment* [25] (f. 11r. Critical transcription in [24]).

The justification of the equality of impact because of the equality of pressure is of course a *petitio principii*, as the principle is justified with the principle itself. However, Newton seemed to think this not problematic, as clearly expressed by this short sentence: "If *r* presse *p* towards *w*, then *p* presseth *r* towards *v*. Tis evident without explication" [25] (f. 13r. Critical transcription in [24]). Whether or not Newton justified the equality of pressure as an evident axiom is discussed in [6].

Until now, action and reaction have somehow been associated with forces, if not coinciding with them. However, there is another way the term action is used by Newton, which is that when, at the end of the scholium added to the section of the laws of motion, he considered the equilibrium of the (simple) machines of pre-Galilean mechanics. In

this case, a modern reader is led to identify action with virtual work, "For if the action of an agent is measured by its force and velocity together, and if, similarly, the forces of resistance arising from their friction, cohesion, weight are measured together with velocity, the action and reaction will always be equal to each other in all examples of using devices or machines" [13] (p. 17). Here, action is measured by the product of force and (virtual) speed, today known as virtual work or virtual power, and the equality of virtual works of contrasting forces is a criterion of equilibrium according to the principle of virtual work [26].

This way of conceiving action as virtual work is also seen from another perspective [1]. That is, in this case, action and reaction concern the equilibrium of a single body, when instead, it is clear in the statement of the PA that they should refer to different bodies. The criticism is unjustified, however, in my opinion. For instance, in the case of a balance, the simple machine that Newton referred to, the two weights $p$ and $q$ suspended at its ends are equivalent if inversely proportional to their (virtual) speeds. This is the equilibrium criterion of the balance considered as a body, a beam. However, one can also see the weight $p$ as the power that exerts an action on the weight $q$ and receives a counter-reaction; the beam here is not considered as a body but simply as a means to transmit forces.

For Newton, action and reaction in this context remain still valid even when the action of the force overcomes the virtual work of resistance. In this case, the machine starts moving with an accelerated motion, and the acceleration gives rise to a resistance, which, added to the resistance of forces, equals the action. Here, one can see at play what is known as d'Alembert's principle, where one considers the force of inertia as a true force [27]. Newton claimed, however, he was not interested in this kind of consideration [13] (p. 17).

The PA has been used in the proof of some propositions related to more than one body, referred to in Appendix A. From the point of view of the theoretical mechanics, the more interesting usage is probably that concerning corollaries 3 and 4 to the laws of motion. Here, Newton concluded that in certain respects, a system of bodies behaves like a single body: "Therefore, the law is the same for a system of several bodies as for a single body with respect to perseverance in a state of motion or of rest. For the progressive motion, whether of a single body or of a system of bodies, should always be reckoned by the motion of the center of gravity" [13] (p. 20).

In the astronomical context, PA is used to study the motion of more than a single planet. Moreover, the universal law of gravitation has been the inspiration for the formulation of the PA, which, in turn, justifies it. For this purpose, the following letter to Cotes, contemporary to the publication of the second edition of the *Principia*, is very interesting:

> *Now the mutual and mutually equal attraction of bodies is a branch of the third Law of motion* [emphasis added] and how this branch is deduced from Phaenomena you may see in the end of the Corollaries of ye Laws of Motion, pag. 22. If a body attracts another body contiguous to it and is not mutually attracted by the other: the attracted body will drive the other before it and both will go away together wth an accelerated motion in infinitum, as it were by a self moving principle, contrary to ye first law of motion, whereas there is no such phaenomenon in all nature [28] (vol. 5. Letter To Cotes, 28 March 1713, p. 397).

## 3. History and Prehistory of the Principle

The central idea of the principle of action and reaction, according to which every action is followed by a reaction, has very ancient roots for the common person, the philosopher, and the mathematician. However, before Newton, only natural philosophers devoted efforts to some understanding of the principle, not only in mechanics [3].

Aristotle did not use the pair of words action and reaction but spoke of reciprocal actions. He did it, for example, in the *Physica* when discussing a body that sets another body into motion: "For to act on the movable as such is just to move it. However, this it does by contact, so that at the same time it is also acted on. Hence, we can define motion as

the fulfillment of the movable qua movable, the cause of the attribute being contact with what can move so that the mover is also acted on" [29] (III.2, 202a).

In the Middle Ages, references to reciprocal action were tacitly abandoned, and the interaction between two bodies or beings was often no longer considered reflexive. One of the subjects had a logical and ontological priority over the other. The stronger (more active) acts, while the weaker (less active) reacts. However, in most cases, the reaction was simply a passion; that is, the weaker subject did not act on the stronger one. The Aristotelians of the 16th century reverted to Aristotelian terminology, but some of them still maintained the difference between action and reaction. Additionally, independently of the use of the word reaction, most of them thought that nothing can resist except by acting upon that which acts upon it; this is at least what the Spanish physician Fancisco Valles thought (1524–1592) [3] (pp. 28–29).

In mechanics, all instances of forces suggested by Newton in the statement of the PA were known since ancient times, with the idea of action and reaction developing differently for each of them. Of course, the first situation, where action and reaction were recognized, concerned pressure, even though the pair of names action/reaction was established later. Indeed, the term *reaction* was not part of classical Latin before the 12th–13th centuries [30] (p. 31).

Actions at a distance were also known very early, for instance, those due to electrified or magnetic bodies. Additionally, celestial bodies were supposed to act on the Earth. There have been essentially two ways of justifying forces at a distance. The oldest justification considers only similar bodies, like attracts like. Additionally, there is a psychological or animistic explanation, the two bodies recognize and spontaneously tend toward each other. Another justification, of a physical kind, admits that there is an external force that acts on a body and makes it move; in such a case, one speaks properly of attraction. While the psychological explanation involves mutual action, the mechanistic one does not necessarily; the attraction may be one-way, in which A may attract B but B does not necessarily attract A.

William Gilbert (1540–1603), in their *De magnete* of 1600, analyzed both electricity and magnetism. In the first case, he spoke of attraction due to a material effluvium emanating from the bodies as a result of rubbing. The attraction is only one-way interaction, for him, where the electrified body attracts small pieces of matter but the latter does not attract the former. In the second case, the cause cannot be found in an effluvium, because magnetism also acts through an obstacle, and the only way Gilbert found this is some form of animism (*coitio*) [31] (p. 97). Some years later, Boyle showed that electricity also has the property of reciprocity [2] (p. 43).

A modern reader finds it difficult to accept any justifications, either psychological or physical; the former because in modern mechanics, at least in classical modern mechanics, there is no room for psychology, and the latter, because at the moment there are no shared explanations, even though in the 17th and 18th centuries, many explanations of the mechanist type were proposed by the likes of Descartes, Johann Bernoulli, and Huygens.

The impact between bodies became the paradigm of mechanics and mechanism of the 17th and 18th centuries. The first studies, such as those of Descartes, were phenomenological or, if one prefers, kinematic. The result of the consequences of the impact, the conservation of momentum, is given without attempting a causal explanation. A different approach is that of Alfonso Borelli (1608–1679), who introduced dynamical considerations for the impact of two bodies A and B that meet with velocities inversely proportional to their bulks:

> Since the impulsive forces of the bodies A and B are equal to each other (because their sizes and velocities are inversely proportional to each other), both bodies therefore strike with equal energy, and both suffer from the other repulses of equal energy since the strength of resistance of body B exactly equals the impulse of A itself, and therefore the direct progress of the body A toward G is altogether impeded by the force of stability [firmitudinis] or resistance of B [32] (p. 120. Translation in [33]).

A not very dissimilar position is reported by John Wallis (1616–1703) in their *Mechanica, sive de motu, tractatus geometricus* of 1670. In [3], some discussions of philosophers of nature discussing impacts are referred to. A certain relief is attributed to Thomas White (1593–1676), an English Roman Catholic priest; White was not, however, read by the mathematicians of the time, and he was probably unknown to Newton himself.

From the brief history given above, it can be concluded that Newton clarified a principle that has always been known, stating that:

1.  Action and reaction are always coupled, they are equal and opposite; when it comes to forces, they act on the same line of action.
2.  Action and reactions are equal (and opposite) not only numerically but also onto-logically, in the sense that to a pressure there is always a pressure as a reaction, to a gravitational force another gravitational force, to an electric force an electric force, etc.

The problem of whether it is correct to use the terms action and reaction in the interaction between bodies persists today in modern mechanics. Newton himself was quite ambiguous in the statement of their principle. While at the very beginning he used the two words *action* and *reaction*, immediately after, he qualified both as actions: "in other words, the actions of two bodies upon each other are always equal and always opposite in direction".

In the case of a collision, for instance, it is difficult to make a distinction between the role of the two bodies; that is, it is difficult to say which acts and which suffers, especially if the two bodies are equal. In the case of forces at a distance (gravitational, magnetic, electric, or otherwise), a body A attracts a body B, which, in turn, attracts A. However, this is not truly a reaction, because the attraction of B is not influenced by the attraction of A but only by its presence. Additionally, indeed, while the terms action and reaction are still sometimes used, these forces are more often referred to as mutual actions.

## 4. Theoretical and Experimental Justifications

In Newton's epistemology, the laws of motion are "deduced" from experience. This also applies to the third law of motion, that is, the principle of action and reaction. Of the three instances of force listed after the statement of the PA, pressure, impact, and gravity, only the second and third instances are justified by Newton, while the first, perhaps the most evident one, is not discussed.

The validity of the principle of action and reaction is verified for the impact experimentally by confirming the conservation of momentum. Newton referred to the experiences of Wren and Huygens for elastic bodies and Wallis for hard (perfectly plastic) bodies [34] (pp. 230–235). However, he carried out experiments by himself considering the collision of bodies arranged at the ends of two pendulums made with wires, along the lines of what was performed by Edme Mariotte [35]. Newton's experiments were very accurate and also took the air resistance into account; the bodies subjected to collisions had a varied nature: "The experiment just described works equally well with soft bodies and with hard ones, since surely they do not in any way depend on the condition of hardness" [13] (p. 24). Newton suggested a theoretical justification already in their commentary on their law. The conservation of momentum derives from the equality of the pressures exchanged by the impacting bodies.

Additionally, for forces at a distance, Newton presented both empirical and theoretical justification. The empirical justification of the equality between action and reaction was carried out by considering the interaction between a loadstone and a piece of iron. It was observed that if two objects are placed in separate vessels that touch each other and float side by side in still water, neither will drive the other forward, which implies the equality of the attraction they will sustain in both directions. The empirical justification was preceded by a theoretical one. Suppose that the loadstone A and the iron B that attract each other are separated by an obstacle to impede their coming together. If A is more attracted toward B than B toward A, then the obstacle will not remain in equilibrium. The stronger pressure will prevail and will make the system of the two bodies and the obstacle move, which is

absurd and contrary to the first law of motion. Notice that Newton also allows the first law of motion to be valid for an aggregate of bodies, even though it is stated for a single body.

However, the most famous justification of the PA is the one concerning gravitational forces. In the scholium to the laws of motion, Newton argued for the equality of the mutual forces of attraction, making it absurd to reason concerning the revolution motion of the earth.

> Let the earth FI be cut by any plane EG into two parts EGF and EGI [Figure 1]; then their weights toward each other will be equal. For if the greater part EGI is cut into two parts, EGKH and HKI, by another plane HK parallel to the first plane EG, in such a way that HKI is equal to the part EFG that has been cut off earlier, it is manifest that the middle part EGKH will not preponderate toward either of the outer parts but will, so to speak, be suspended in equilibrium between both and will be at rest. Moreover, the outer part HKI will press upon the middle part with all its weight and will urge it toward the other outer part EGF, and, therefore, the force by which EGI, the sum of the parts HKI and EGKH, tends toward the third part EGF is equal to the weight of the part HKI, that is, equal to the weight of the third part EGF. Additionally, therefore the weights of the two parts, EGI and EGF, toward each other are equal, as I set out to demonstrate. Additionally, if these weights were not equal, the whole earth, floating in an aether free of resistance, would yield to the greater weight and in receding from it would go off indefinitely [13] (p. 25. Translation into English in [21]).

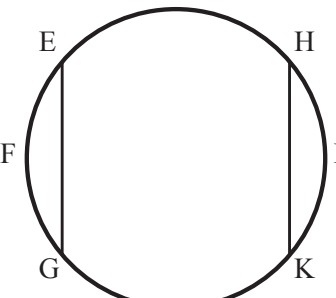

**Figure 1.** The principle of action and reaction for centripetal/gravitational forces (Redrawn from [13], p. 25).

In the above justifications of the PA, the first and second laws of motion were used, together with experimental evidence. However, it must be said that Newton's three laws of motion are closely interrelated [36] (Chapter 7). For example, it is possible to deduce, albeit with some freedom, the second law of motion from the PA in at least two ways.

In the first way, the second law of motion can be justified based on the evidence of the conservation of momentum in the impact between two bodies A and B and the PA, according to which in the impact, the two bodies exchange equal pressures and, therefore, are subject to the same impressed force ($f_A = -f_B$; Def. 4). If the impressed forces are the same and the variations in the momentum are the same as well, ($m_A \Delta v_A = -m_B \Delta v_B$), this means that the impressed force is equal to the variation in the momentum: $f_A = m_A \Delta v_A$ and $f_B = m_B \Delta v_B$. Thus, the second law of motion is verified.

In a second way, one resorts to an ambiguity that persists in *Principia*, as indeed in all of Newton's writings, between inertia as force and inertia as mass. For example, in the commentary on the definition of vis insita, Newton wrote:

> *Definition 3. Inherent force of matter* [...]. This force is always proportional to the body and does not differ in any way from the inertia of the mass except in the manner in which it is conceived. Because of the inertia of matter, every body is only with difficulty put out of its state either of resting or of moving.

Consequently, inherent force may also be called by the very significant name of force of inertia. Moreover, a body exerts this force only during a change of its state, caused by another force impressed upon it, and this exercise of force is, depending on the viewpoint, both resistance and impetus; resistance insofar as the body, in order to maintain its state, strives against the impressed force [13] (p. 2. Translation into English in [21]).

Newton's text can be read easily as if a body reacts with a force to a change in motion. The force can be named as inertia force $f_I$; and the reaction is the greater the greater the mass and the acceleration; so it measured by $f_I = -ma$. The impressed force $f$ for the PA is equal and contrary to force of inertia, that is $f_i = -f$; it is thus $f = ma$, that is, the second law of motion (a modern version indeed).

Another point of view suggesting the interrelation among Newton's laws of motion is offered by Clifford A. Truesdell as an interesting attempt to axiomatize classical mechanics in [37], as shown in Appendix B.

## 5. Reactions of Constraints

Of the examples Newton gave after the statement of PA, there is that of the pressure of a finger on a stone, in which the finger in turn undergoes an equal and opposite pressure from the stone. Given the dynamic approach of the *Principia*, Newton did not dwell on this aspect, which is instead fundamental in statics. Put otherwise, if one considers the stone as a constraint, Newton was saying that constraints are capable of exerting forces that are the dynamic counterpart of the geometric characterization of the constraint. This is a substantially new position, which was not accepted by scholars of statics of the time, for whom constraints were anything but an agent capable of providing forces.

Indeed, the core from which the idea of force developed in mechanics is to be found in humans' awareness of their own physical efforts necessary to interact with the outside world, to move bodies for instance. This idea was later given a name, which, through etymological and conceptual transformations, became the term *forza* in the Italian language, *force* in English and French, and *Kraft* in German. It is no coincidence that physical effort is the only concept of force that the majority of people still clearly possess today. Given this conception of force as something that requires activity, it is nearly impossible to conceive forces as coming from inanimate bodies, as constraints are assumed to be.

The idea of force as physical effort is more than sufficient to establish the foundations of statics, the discipline that studies bodies in equilibrium. Physical efforts can be measured with weights, to which it is in principle neither useful nor necessary to assign any ontological status. Weights exist and did not give particular problems to the scientists of the past. With weights, it was then possible to measure actions other than those of physical effort, such as, for example, those induced by stretched or compressed springs.

The case of the interaction of bodies with constraints, which are considered uncritically only as geometric entities, explains the use of the action/reaction pair since the force exerted by the constraint and that exerted on the constraint seem to be ontologically different (somehow, therefore, strictly speaking, the principle of action and reaction according to which the two forces must be of the same type is violated). The force acting on the constraint, for example, that due to a compressed spring that is brought close to the constraint, has all the characteristics of the normally admitted forces, and it has its own well-defined value measured by the amount of the contraction of the spring; in rational mechanics, this type of force is qualified as *active*. The force exerted by the constraint originates only when the active force acts on the constraint, and for this reason, it is called *passive* or reactive. It is always equal and opposite to the active force and is potentially unlimited; that is, the constraint can exert a reaction of any great intensity.

A more interesting case that concerns the action–reaction principle for constraints, with respect to the finger and the stone, is that of a perfectly rigid horizontal plane on which a heavy body C rests and is in equilibrium. Some modern textbooks proceed as follows, the force of gravity measured by the weight $g$ acts on C; since C is in equilibrium,

the table must exert a reaction $p = -g$ on C. In this way, the weight $g$ is contrasted by the reaction $-p$.

The analysis is clearly misleading, as the principle of action and reaction requires that action and reaction be applied to different bodies and that they have the same ontological nature. A more correct analysis (as far as possible from the idealized schematization of the constraint), would consider that the weight $g$, which urges C downward, determines a pressure $p = g$ on the table that reacts with another pressure $-p$ acting on C. The action and reaction pair is thus given by terms $p$ and $-p$ and not $g$ and $-p$.

One of the first occurrences in which some activity is given to the constraint was due to Emmanuel Maignan (1601–1676) in their *Perspectiva horaria sive de horographia gnomonica tum theoretica, tum practica* of 1648, where he, differently from the Cartesian tradition, used a dynamic approach to the impact.

> Lemma I. There is a mutual interaction between the incident ball and the plane on which it strikes, i.e., a greater or lesser opposition, in proportion as the incidence of the total motion is more or less perpendicular.
> When a ball LDON falls on to a plane surface AB with some force and impact on it, there is an action of the ball on the plane, that is a percussion. Additionally, since whatever acts on another suffers from it in turn, the ball *suffers something* [emphasis added] from the plane; and so there is a reaction of the plane on the ball, that is a re-percussion [38] (p. 293).

Notice that Maignan did not, however, speak of the force of the table on the ball; he just said that the ball *suffers something* from the table.

Notice that Maignan did not, however, speak of the force of the table on the ball; he just said that the ball *suffers something* from the table. No 16th and 17th century scholars of statics made explicit use of reaction as a force. This is also true of one of the most authoritative treatises on statics of the 18th century based on a mature concept of force (obeying the law of the parallelogram), that of Pierre Varignon (1654–1722), *La nouvelle mécanique ou statique*, published in 1725 after Varignon's death but conceived much earlier [39]. In all cases, the forces were only the active ones and derive from muscular actions or weights. Very frequently, the forces are presented in drawings as hands that lift objects directly or by means of ropes, as shown in Figure 2.

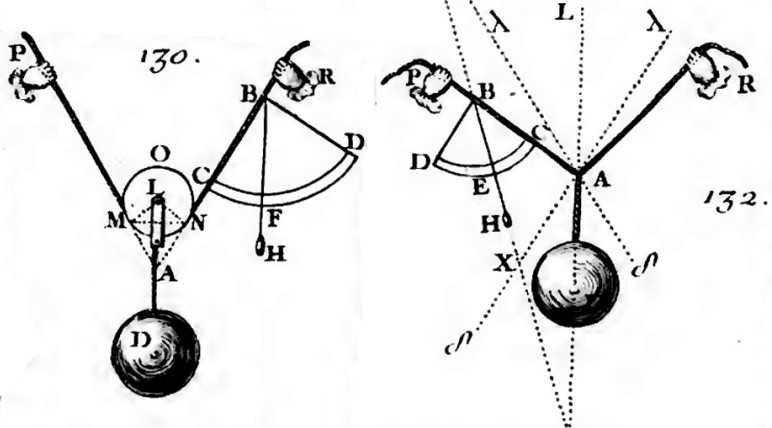

**Figure 2.** Example of muscle force [39], vol. 1, p. 270.

To understand the novelty introduced by Newton, even without the particular attention he paid to it, it is sufficient to consider how reactions were treated in Newton's time. For instance, Descartes, referring to the balance of the pulley, wrote:

> THE PULLEY.—Let ABC be a rope passing around the pulley D [Figure 3], in which the weight $E$ is applied and suppose first that two men support or raise equally each of the two ends of the rope, it is clear that if the weight weighs

200 pounds, each of the two men take, to support or lift, the force required to support 100 pounds, because each holds only one half. Let then that A, one end of the rope, being attached to a nail, the other C is still supported by a man, it is clear that this man in C would need, as before, only the force required to support 100 pounds, because the support is to do the same service of the man who was supposed before [40] (Letter to Constantine Huygens 5 October 1637, vol. 1, p. 437).

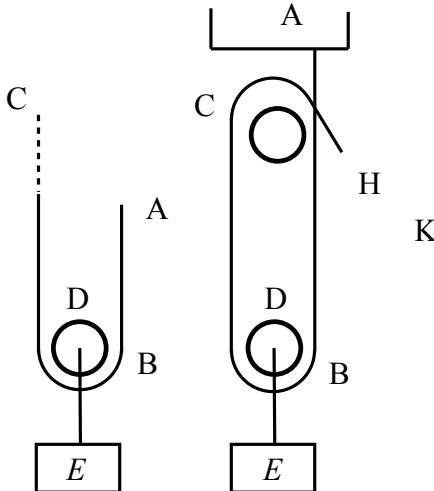

**Figure 3.** The pulley. Redrawn from [40] (Letter to Constantine Huygens 5 October 1637, vol. 1, p. 437).

Note the rhetorical device used to overcome the conceptual difficulty connected with the constraint reactions. First, Descartes considered two equal muscular forces applied to the extremes C and A of the rope that supports the pulley of weight E in Figure 3. He then replaced a force with a constraint; that is, he admitted that the side A of the rope is attached to a support and declared that things do not change. In this exhibit, he demonstrates that 1/2 E of muscle force is needed to lift E.

A most interesting example is due to Johann Bernoulli related to the situation of Figure 4. The goal was to find the *impression* that each of the two inclined planes CA and CD receives from the ball. Bernoulli determined an *impression* at a time, thinking of replacing one of two planes with a force, for example, the plane CD with a force *R* orthogonal to it. This is the classic approach in statics to replace a constraint with forces and apply the rules of equilibrium to the resulting system when the body is not constrained. Bernoulli applied their principle of energies to evaluate *R* [41] (Letter from Johann Bernoulli to Pierre Varignon, 26 February 1715).

Unfortunately, we do not know what Newton might have thought about the table supporting a heavy body. One can imagine that if he had concentrated on the problem of constraints, he would have avoided their extreme schematization and would have considered them as bodies formed by corpuscles that exert an action at a distance.

For example, Newton could have argued as follows, as the body C approaches the plane, the forces acting at a distance on its particles change from attractive to repulsive so as to balance the gravity acting on C (see Figure 5). In turn, C interacts with the particles of the body, resulting in a state of compression.

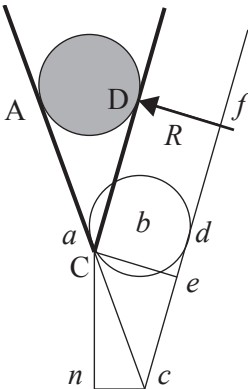

**Figure 4.** The impression on a support. Redrawn from [41] (Letter from Johann Bernoulli to Pierre Varignon, 26 February 1715).

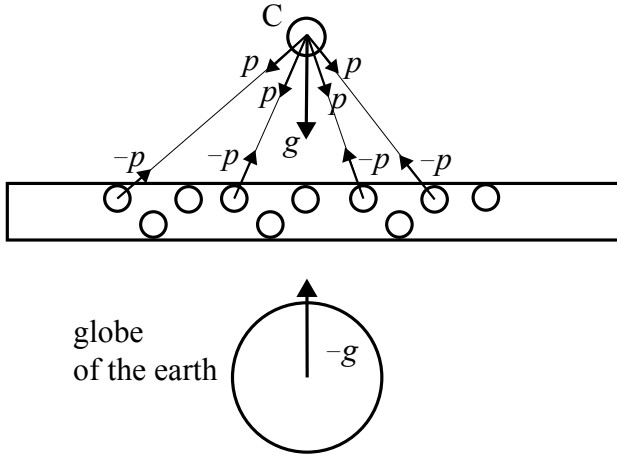

**Figure 5.** Action and reaction for a support of a heavy body.

### 6. Conclusions

The principle of action and reaction in its simplest formulation, i.e., that every action is followed by a reaction, has very ancient roots, but it is only in the 17th century, with Newton, that it assumed a precise and quantitative formulation reaching the dignity of the third law of motion. Although Newton considered their third law very important, to the point of assuming it as the only law that was their own contribution, it has been the object of relatively little attention by historians of science, who focused on the first two laws, the principle of inertia and the second law of dynamics. In reality, PA is very important at a foundation level; more than the definition of the impressed force, it deals with the ontology of force, differentiating it as a cause, an essentially qualitative concept belonging to natural philosophy, from its action, which becomes a mathematical concept endowed with a measure.

PA finds its definitive formulation in the *Principia*, in a more or less unchanged form in all the three editions, but it can also be found in Newton's previous works that highlight its genesis. Newton stated PA in a quite ambiguous statement, at least for a modern theorist. To avoid the use of the term *force*—which was a dangerous subject—he, also referring to tradition, used the not less dangerous term *action*, in the following form: to any action there is always an opposite and equal reaction. In other words, the actions of two bodies upon each other are always equal and always opposite in direction.

Despite its simplicity and apparent evidence, Newton felt the need to submit PA to both experimental and theoretical proof, which are both considered fairly convincing.

**Funding:** This research received no external funding.

**Data Availability Statement:** No new data were created or analyzed in this study. Data sharing is not applicable to this article.

**Conflicts of Interest:** The author declares no conflict of interest.

## Appendix A. Use of PA in the *Principia*

In this appendix the propositions where PA was used explicitly in the third edition of the *Principia* [13] are referred to, with the page of occurrence. PA is cited either as Third law (*Lex tertia*) or as Law 3 (*Lex 3*). The translation into English is due to Cohen and Withman [21].

p. 17 **Corollary 3 to the laws of motion**. *The quantity of motion, which is determined by adding the motions made in one direction and subtracting the motions made in the opposite direction, is not changed by the action of bodies on one another.*

FOR AN ACTION AND THE REACTION OPPOSITE TO IT ARE EQUAL BY LAW 3, and thus by law 2, the changes which they produce in motions are equal and in opposite directions.

p. 160 **Section XI. Book I**. *The motion of bodies drawn to one another by centripetal forces.*

Up to this point, I have been setting forth the motions of bodies attracted toward an immovable center, such as, however, hardly exists in the natural world. For attractions are always directed toward bodies, AND—BY THE THIRD LAW—THE ACTIONS OF ATTRACTING AND ATTRACTED BODIES ARE ALWAYS MUTUAL AND EQUAL; so that if there are two bodies, neither the attracting nor the attracted body can be at rest, but both (by corol. 4 of the laws) revolve about a common center of gravity.

p. 187 **Proposition 69. Book I**. *If, in a system of several bodies A, B, C, D, . . . , some body A attracts all the others, B, C, D, . . . , by accelerative forces that are inversely as the squares of the distances from the attracting body; and if another body B also attracts the rest of the bodies A, C, D, . . . , by forces that are inversely as the squares of the distances from the attracting body; then the absolute forces of the attracting bodies A and B will be to each other in the same ratio as the bodies [i.e., the masses] A and B themselves to which those forces belong.*

For, at equal distances, the accelerative attractions of all the bodies B, C, D, . . . toward A are equal to one another by hypothesis; and similarly, at equal distances, the accelerative attractions of all the bodies toward B are equal to one another. Moreover, at equal distances, the absolute attractive force of body A is to the absolute attractive force of body B as the accelerative attraction of all the bodies toward A is to the accelerative attraction of all the bodies toward B at equal distances; and the accelerative attraction of body B toward A is also in the same proportion to the accelerative attraction of body A toward B. However, the accelerative attraction of body B toward A is to the accelerative attraction of body A toward B as the mass of body A is to the mass of body B, because the motive forces—which (by defs. 2, 7, and 8) are as the accelerative forces AND THE ATTRACTED BODIES JOINTLY—ARE IN THIS CASE (BY THE THIRD LAW OF MOTION) EQUAL TO EACH OTHER. Therefore the absolute attractive force of body A is to the absolute attractive force of body B as the mass of body A is to the mass of body B. Q.E.D.

p. 194 **Proposition 75, Corollary 2. Book 1**. The same is true when the attracted sphere also attracts. For its individual points will attract the individual points of the other with the same force by which they are in turn attracted by them; and thus, since in every attraction THE ATTRACTING POINT IS AS MUCH URGED (BY LAW 3) AS THE ATTRACTED POINT, the force of the mutual attraction will be duplicated, the proportions remaining the same.

p. 239 **Section 1, Scholium. Book 2**. For by the action of a swifter body, a motion that is greater in proportion to that greater velocity is communicated to a given quantity of the medium in a smaller time; and thus in an equal time, because a greater quantity of the medium is disturbed, a greater motion is communicated in proportion to the square of the velocity, AND (BY THE SECOND AND THIRD LAWS OF MOTION) the

resistance is as the motion communicated. Let us see, therefore, what kinds of motions arise from this law of resistance.

p. 283    **Section 5, Case 3. Book 2**. I say furthermore that there is equal pressure on different spherical parts. FOR CONTIGUOUS SPHERICAL PARTS PRESS ONE ANOTHER EQUALLY IN THE POINT OF CONTACT, BY THE THIRD LAW OF MOTION. However, by case 2, they are also pressed on all sides by the same force. Therefore any two noncontiguous spherical parts will be pressed by the same force, since an intermediate spherical part can touch both. Q.E.D.

p. 283    **Section 5, Case 4. Book 2**. I say also that all the parts of the fluid are equally pressed on every side. For any two parts can be touched by spherical parts in any points, and there they press those spherical parts equally, by case 3, and in turn are equally pressed by them, by the third law of motion. Q.E.D.

p. 322    **Section 7. Corollay 5. Book 2**. Additionally, since similar, equal, and equally swift bodies, in mediums which have the same density and whose particles do not recede from one another, impinge upon an equal quantity of matter in equal times (whether the particles are more and smaller or fewer and larger) and IMPRESS UPON IT AN EQUAL QUANTITY OF MOTION AND IN TURN (BY THE THIRD LAW OF MOTION) UNDERGO AN EQUAL REACTION FROM IT (that is, are equally resisted), it is manifest also that in elastic fluids of the same density, when the bodies move very swiftly, the resistances they encounter are very nearly equal, whether those fluids consist of coarser particles or are made of the most subtle particles of all. The resistance to projectiles moving very quickly is not much diminished as a result of the subtlety of the medium.

p. 341    **Section 7, Lemma 6. Book 2**. With the same suppositions, these bodies are equally urged by the water flowing through the channel. THIS IS EVIDENT BY LEM. 5 AND THE THIRD LAW OF MOTION. Of course, the water and the bodies act equally upon one another.

p. 357    **Section 8, Corollary. Book 2**. If some part of a pressure propagated through a fluid from a given point is intercepted by an obstacle, the remaining part (which is not intercepted) will spread out into the spaces behind the obstacle. This can be proved as follows. From point A let a pressure be propagated in any direction and, if possible, along straight lines; and by the obstacle NBCK, perforated in BC, let all the pressure be intercepted except the cone-shaped part APQ, which passes through the circular hole BC. By transverse planes *de*, *fg*, and *hi*, divide the cone APQ into frusta; then, while the cone ABC, by propagating the pressure, is urging the further conic frustum *degf* on the surface *de*, and this frustum is urging the next frustum *fgih* on the surface *fg*, and that frustum is urging a third frustum, and so on indefinitely, OBVIOUSLY (BY THE THIRD LAW OF MOTION) the first frustum *defg* will be as much urged and pressed on the surface *fg* by the reaction of the second frustum *fgfhi* as it urges and presses the second frustum.

p. 399    **Proposition 5, Corollary 1. Book 3**. Therefore, there is gravity toward all planets universally. For no one doubts that Venus, Mercury, and the rest [of the planets, primary and secondary,] are bodies of the same kind as Jupiter and Saturn. ADDITIONALLY, SINCE, BY THE THIRD LAW OF MOTION, EVERY ATTRACTION IS MUTUAL, Jupiter will gravitate toward all its satellites, Saturn toward its satellites, and the earth will gravitate toward the moon, and the sun toward all the primary planets.

p. 404    **Proposition 7. Book 3**. *Gravity exists in all bodies universally and is proportional to the quantity of matter in each.*
Further, since all the parts of any planet A are heavy [or gravitate] toward any planet B, and since the gravity of each part is to the gravity of the whole as the matter of that part to the matter of the whole, AND SINCE TO EVERY ACTION (BY THE THIRD LAW OF MOTION) THERE IS AN EQUAL REACTION, it follows that planet B will gravitate in turn toward all the parts of planet A, and its gravity toward any one part will be to its

gravity toward the whole of the planet as the matter of that part to the matter of the whole. Q.E.D.

**Appendix B. Formal Comparison between the Principle of Inertia and PA**

In the section dealing with definition and properties of forces of *A first course in rational continuum mechanics* by Clifford Ambrose Truesdell [37], some axioms very intuitive at least for a mathematician, are introduced. In particular force, a primitive term of the theory, is considered as an additive magnitude, that is:

Axion F2.     $\mathbf{f}(\mathcal{C}_1 \vee \mathcal{C}_2, \mathcal{B}) = \mathbf{f}(\mathcal{C}_1, \mathcal{B}) + \mathbf{f}(\mathcal{C}_2, \mathcal{B})$ for any $\mathcal{B}, \mathcal{C}_1, \mathcal{C}_2 \in \Omega$

Axion F3.     $\mathbf{f}(\mathcal{B}, \mathcal{C}_1 \vee \mathcal{C}_2 = \mathbf{f}(\mathcal{B}, \mathcal{C}_1) + \mathbf{f}(\mathcal{B}, \mathcal{C}_2)$

where $\mathcal{C}_1, \mathcal{C}_2, \mathcal{B}$ are distinct bodies, $\mathbf{f}(\mathcal{C}, \mathcal{B})$ is the force of $\mathcal{B}$ on $\mathcal{C}$, $\Omega$ is the set of all bodies and $+$ is the ordinary sum between vectors [37] (p. 20).

As the property of additivity of force implies the law of parallelogram, Truesdell in substance is considering it as an axiom, while for Newton the rule of parallelogram is a theorem deduced from the laws/axioms of motion.

If the force exerted by $\mathcal{C}$ on $\mathcal{B}$ is of magnitude equal and of sign opposed to that exerted by $\mathcal{B}$ on $\mathcal{C}$, that is:

$$\mathbf{f}(\mathcal{B}, \mathcal{C}) = -\mathbf{f}(\mathcal{C}, \mathcal{B}) \quad \forall \mathcal{B}, \mathcal{C} \in \Omega \tag{A1}$$

the system of forces $\mathbf{f}$ is said to be *pairwise equilibrated* [37] (p. 21).

The following theorem is proved:

> Theorem (Noll, Gurtin, and Williams). A system of forces is pair-wise equilibrated if, and only if, the resultant forces $\mathbf{f}(\mathcal{B}, \mathcal{B}^{\mathbf{e}})$, regarded as a function of $\mathcal{B}$, is additive on the separate bodies of $\Omega$ [37] (p. 21).

where $\mathcal{B}^e$ is the set of bodies other than $\mathcal{B}$.

The result is described as follows:

> In the past, instances of (6) were often inferred from a vague "axiom" called the law of "action and reaction", which was regarded as expressing the content of Newton's Third Law of Motion: "To an action there is always a contrary and equal reaction; or, the actions of two bodies mutually upon one another are always equal and directed toward contrary parts." *If, indeed, what NEWTON meant by "action" is what we here call "force", which is by no means clear from their own words or the contexts in which he applied them* [emphasis added], then the above argument shows that axiom to be equivalent, as far as pairs of separate bodies are concerned, to additivity of resultant forces on separate bodies. This fact is independent of whatever relations there may be among forces and motions [37] (p. 22).

A more interesting theorem, from the point of view of the present entry, on pairwise equilibrated forces is the following one:

Theorem (Rizzo). In order that:
$\mathbf{f}(\mathcal{B}, \mathcal{C}) + \mathbf{f}(\mathcal{C}, \mathcal{B}) = \mathbf{0}$ for any $\mathcal{B}, \mathcal{C} \in \Omega$
it is necessary and sufficient that:
$\mathbf{f}(\mathcal{B}, \mathcal{B}) = \mathbf{0}$ for any $\mathcal{B} \in \Omega$ [37] (p. 28).

The term $\mathbf{f}(\mathcal{B}, \mathcal{B})$ is known as the *self-force*, and represents the force that a body exerts on itself. It is clear that in classical mechanics this force should be zero, otherwise the principle of inertia does not hold. So Truesdell's axiomatic suggests a logical link between the principle of inertia and PA.

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
