# Peer review of "The Principle of Action and Reaction According to Newton"

_encyclopedia, doi:10.3390/encyclopedia3020051_

Round 1
Reviewer 1 Report
This article contains a series of reflections on Newto's so-called third law, or the law of action and reaction. The author examines these concepts historically (from Aristotle to the Middle Ages) and concludes that it is in the Principia that Newton establishes the subject in an essential way. The euphemism used by Newton for action in order not to use the concept of force is discussed. This third law has been less studied than the first two. The author reviews its impact on science and philosophy up to the present day. It is a very interesting piece of work that I enthusiastically recommend to accept.
Author Response
Thanks for your comments.
English has been improved and the template of MDPI has been used
Reviewer 2 Report
I think the paper is very well written. It is a historical study from the history of physics, which is not prepared only from the point of view of the glorification of figures who are today considered great in this field. As long as the historian of science does not recognize the cumulative development of science, but at least implicitly shares the ideas of personalities like Laudan, Koyré, Kuhn, Lakatos, we consider it good. We believe that the author plastically expressed the positive but also contradictory opinions of the time. We positively appreciate that the authors elaborated the history and prehistory of the described principle. We consider the physical analysis of the described phenomenon to be sufficient. The contribution is written critically, also with the expression of those aspects that Newton as a physicist handled less brilliantly. The structure of the text is good. The authors also touch on etymology, which should also be positively appreciated. The number of bibliographic references is relatively low, we recommend increasing it. We also recommend increasing the number of sources by several publications of recent provenance (not older than five years). The conclusion is sufficient and concise. A lot of important information is contained in the appendices.
It is also necessary to comment on the fact that some paragraphs of the paper are not original, but were copied from other sources. These are mainly sources: Isaac Newton. "BOOK 2: THE MOTION OF BODIES", University of California Press, 2019, www.hrstud.unizg.hr, Isaac Newton. "BOOK 1: THE MOTION OF BODIES",
University of California Press, 2019 and others. In those places where the texts are copied, it is necessary to delete the passages and replace them with new text. Our verdict is: big changes. We ask the authors to correct the mentioned shortcomings. After subsequent corrections, the text can be published.
Author Response
Thanks for your comments and suggestions.
English has been improved and the template of MDPI has been used.
Regarding the references, I added some regarding experimentation and pedagogy, and a few on foundation. For the parts copied from other sources, I must say that they only regard translations of Newton's writings. They all come from: "Newton, I. The Principia. Mathematical principles of natural philosophy. Translated into English by 666 Cohen IB, Withman A (assisted by Budenz J); University of California Press: Oakland", 1999, which is an early edition of the book from University California Press 2019, you cite.
I could translate Newton Latin by myself, but my English is not as good as that of Cohen and Withman.
Reviewer 3 Report
This is an informative entry for the encyclopedia but unfortunately, there are too many linguistic mistakes, which need to be corrected. Many of the footnotes could go if the page numbers were included in the square brackets: [22, p. 21]. On p. 7 the sentence: 'The theoretical justification was preceded by a theoretical one' does not make sense to me. On the same page the first sentence of the Newton quote ('Let the earth...') is incomplete. All the other quotes should be checked again.
For an entry to an encyclopedia, the quality of English is unacceptable. The author should ask a native speaker to go through the manuscript and correct all the linguistic mistakes.
Author Response
Thanks for your comments.
Thanks for having signaled two meaningful typos. Of course all the quotations have been revised.
English has been edited by the MDPI english-editing service, and I hope it has been improved.
For what footnotes and citations are concerned, I changed them according to the journal prescriptions and I used MDPI template; I hope they appear clearer.
Round 2
Reviewer 2 Report
I believe that the paper can be accepted in the fashion presented by the authors. The paper was supplemented in many ways, including new bibliographic sources.
Reviewer 3 Report
Please correct two remaining mistakes:
- p. 7, line 327: In In...
- p. 8, line 360: Kraft